# OpenReview forum: "TherapyGym: Evaluating and Aligning Clinical Fidelity and Safety in Therapy Chatbots"
_ICLR.cc/2026/Conference — Submitted to ICLR 2026_

### Official Review · Reviewer_RSRq · 2025-10-30

**Soundness:** 1
**Presentation:** 2
**Contribution:** 2
**Rating:** 2
**Confidence:** 4

**Summary:**

This paper present TherapyGym, an evaluation framework to assess two aspects (fidelity and safety) of therapy chatbots. Fidelity means " how skillfully a therapist implements a treatment model" e.g. adherence and competence. Authors adopted  Cognitive Therapy Rating
Scale (CTRS) for this aspect. Safety refers to "requires therapists to avoid harmful behaviors, which in chatbot settings demands additional constraints". Authors captured 4 "complementary suite of tests targeting chatbot-specific risks" to represent the Safety aspect. They simulated multi-turn dialogue between therapist and client to test llms. They also released THERAPYJUDGEBENCH to test different judge models performances. At the end, they did RL finetuning with GRPO to compare the performance differences of models before and after on their evaluation pipeline/human labels.

**Strengths:**

S1 importance of topic.
- the topic of evaluating therapy chatbot is crucial, especially with many news/cases of users being affected negatively after using chatbot for their own psychotherapy session.

S2 Construct a expert-annotated judge dataset
- authors recruited two experts to annotate on their simulated conversations, which could treat as a way to train a better judge model in psychotherapy field.

**Weaknesses:**

W1: Biggest weakness is that there are insufficient validations/measures to ensure that the two components of the evaluation pipeline reflect real-world therapy and clients accurately, and to ensure that the evaluation pipeline’s performance is comparable to that of human experts.

(1) Simulation conversations: the authors claimed the conversations are realistic in line 94 (imo which is crucial for a practically important evaluation) but I doubt it:

- Simulated conversations are not good reflective of the real world theory-client conversations
-  limited to 10 turns while real-world has long conversation in therapy session (usually at least 15mins-1hr).
- did not do any validation on the quality of the conversation (e.g., Consistency score conducted by [1])

Also, authors adopt different model choices for simulation (e.g. using o3-mini to implement the patient), which suggests that authors have some criteria for selection, but the paper provides no any explanation/elaboration. For instance, I would expect some human validations comparing different models simulations on role of patient, along with a cost-performance analysis to justify the choices.

(2) llm judge (TherapyJudgeBench)
- authors did not explain how many experts annotate the tables on the judge bench. I assume authors use the two licensed practitioners labels (as in Table 2). First, the low interrater reliability on some aspects (e.g. \alpha = 0.2 in Guid.) suggests that there may be issues with providing guidance and standardizing the grades from human experts during data collection. I believe this could seriously undermine the evaluation pipeline and all the claimed performance improvements through RL/GRPO, since we are not sure whether the gold labels are correct (here I assume the authors used human labels since they did not state it clearly). Could you share more on how you avoid this issue (see Q3 i)?

- Second, the authors miss details about the practitioners and how they perform annotation. These details are needed to help readers judge whether their labels are trustworthy.

W2 Lack of novelty and not sure the explicit differences compared with prior works.
- the simulation between therapist and clients has been explored since 2023 ([1], [2]). There are many important unsolved follow-ups in the field. For instance, how to ensure the simulated multi-turn conversations are consistent.
- Authors compared different benchmarks in Table 1, but I do not find the comparison is meaningful. First, I do not understand the meaning of each dimensions. (e.g. what is "Skill Granularity" or "safety sensitivity"?). I would suggest authors provide more examples and definitions of each.
- More critically, it feels to me the comparison table is a bit biased. It serves as to justify their evaluation pipeline as novel by merely adding the features/experiments authors implemented as dimensions to compare with prior works. I expect they can walk through the reason why we need to include those dimensions (with analysis e.g. what is lacking from the current simulation approach suggested by others). The motive of "but these typically assess generic traits such as “empathy”, “fluency”, and “helpfulness” without grounding in clinically validated constructs." (line 74-75) is good but I do not see how exactly the dimensions authors suggested relate to what they claim.

W3 lack of many important implementation details, metric details and lack of in-depth discussion
- the writing is rough in discussion section, when compared with the earlier sections. I recommend more in-depth analysis with examples and case studies on the failure modes of models. It could benefit the community more by showing how exactly the models improve (through RL/GRPO) via qualitative analysis and examples.
- several implementation details  (See q2, q4) and metric calculation (See Q6) are missing
- how the metric (safety violation rate in Table 4) is calculated?

[1] https://arxiv.org/abs/2401.00820
[2] https://arxiv.org/abs/2405.19660

**Questions:**

Q1 why select o3-mini to implement patient? (line 198-199)

Q2 how you get patient profiles? (line 354-355)

Q3
i) how you ask clinicians annotate: (table 2)
\alpha = 0.2 is very low and it looks like annotators do not have good guidance/ definitions beforehand
from line 315- 318, there are descriptions on instructing the annotators in different stages. can authors provide more info?
ii) what are their background? how long do they practice in CBT?

Q4 definition of each aspects to help reader interpreting

Q5 table 4 -- authors compared between human vs ai. what is the gold label? human vs human has low consistency

Q6 table 3: why it has nan?

Q7 simulation examples? not sure how good the simulated dialogues are. (W1 (1))

Q8 "Safety is captured through a complementary suite of tests targeting chatbot-specific risks, including judgmental
behavior, failure to address harm and speculation about symptoms." I'm not sure what it means. What are the suite of tests? Do you mean you do literature search and classify some important safety risks related to counselling?  (line 89-90)

minor: fig 3 numbers are being overlapped by the plotted circles.

---

> ### Author Response · Authors · 2025-11-23
>
> Thank you for your valuable feedback.
>
> ## W1 - Validity of simulations and judges
>
> Before addressing the specific points, we want to clarify our stance on simulation. In many safety–critical domains such as *autonomous driving* [2] and *robotics*[3][4], **high-fidelity simulators are not a shortcut but a necessary tool**: they avoid exposing real users to unsafe or immature systems, enable large-scale and repeatable experimentation, and allow systematic stress-testing of rare but important scenarios. **What matters in these settings is not “simulation vs. reality” in the abstract, but whether policies that perform well in simulation also behave well under independent real-world evaluation.** Our work is designed around exactly this principle: we construct a clinically grounded simulation and then explicitly test whether its preferences align with expert ratings.
>
> Our aim is therefore not to claim that our setup fully reproduces real-world therapy, but to build a **scalable, ethically acceptable, clinically anchored** environment with two key properties:
>
> 1. The simulated conversations are plausible CBT-style sessions grounded in validated cognitive models, and
> 2. The environment’s scores track expert judgments well enough to support model comparison and RL.
>
> ### (1) For simulated conversations
>
> *Why simulation and what level of realism we target:* Evaluating therapy agents interactively requires a conversational partner who is either a real patient or a simulator. **Running every evaluation with real patients would be logistically and ethically prohibitive** (risk management, informed consent, repeated exposure to suboptimal models). A carefully designed simulator is thus a practical compromise: **we target clinical plausibility at the level of CBT case formulation and interaction patterns, rather than claiming to capture all human nuances.**
>
> To ground the simulations, we do not ask models to “role-play a depressed person” in the abstract. Instead, we initialize patients from the **Patient-Ψ-CM cognitive models**[1], which encode CBT-relevant constructs (core beliefs, automatic thoughts, situations, emotions, behaviors). This anchors each dialogue in a validated CBT conceptualization. **The Patient-Ψ-CM paper [1] already reports clinician ratings of patient realism, and TherapyGym reuses this pipeline**; we will make this inheritance and the concrete patient instructions explicit. We also acknowledge that phenomena such as strong resistance or highly ambivalent clients are only partially captured, so our environment is a controlled approximation, not a replacement for clinical trials.
>
> *Why 10-turn “micro-sessions”:* We use 10-turn sessions as a standard micro-session that is long enough to exercise multiple core CBT skills (agenda setting, understanding, guided discovery, strategy for change, homework), while keeping expert annotation and RL training tractable. Ten turns roughly correspond to a 10–15 minute focused segment of a therapy session, a unit that is commonly used in supervision and skills training. The field of LLM-based psychotherapy is still in an exploratory phase; compared to prior work that evaluates single turns (CBTBench [5], CounselBench [6], PsychoCounsel [7]), our CBT-anchored 10-turn dialogues already represent a clear step forward in interaction depth. Extending the framework to full 50-minute sessions and multi-session trajectories is an important next step, but is currently constrained by annotation cost and RL stability; our present work explores the feasibility of this intermediate, yet richer, setting.
>
> *Why we do not adopt the “consistency score” the reviewer mentioned from [8]* The real conversations used in [8] are **not CBT-style therapy sessions**, and the proposed “consistency score” is designed to characterize LLM therapist behavior (e.g., stability of stance across turns), rather than to assess validity of simulated patients. In our setting, the main bottleneck for such validation is the lack of publicly available CBT session transcripts: due to privacy and regulatory constraints, **real conversation datasets in psychotherapy are extremely rare**, and to the best of our knowledge there is currently *no public corpus of CBT therapy dialogues* suitable for this purpose. If the reviewer found a CBT conversation dataset, we would be very grateful for a pointer and would be excited to evaluate TherapyGym on it in future work.
>
>
> 1. Wang et al. EMNLP 2024 https://arxiv.org/abs/2405.19660
> 2. Goff et al. 2025 https://arxiv.org/pdf/2504.19077v1
> 3. Hao et al. CVPR 2025 https://arxiv.org/pdf/2504.07095
> 4. Peng et al. ICRA 2018 https://arxiv.org/pdf/1710.06537
> 5. Zhang et al. 2025 NAACL https://arxiv.org/abs/2410.13218
> 6. Li et al. 2025 https://arxiv.org/abs/2506.08584
> 7. Zhang et al. 2025 https://arxiv.org/abs/2502.19731
> 8. Chiu et al. 2024 https://arxiv.org/abs/2401.00820

---

> > ### Author Response · Authors · 2025-11-23
> >
> > ### (2) For the judge
> >
> > The human raters consisted of a licensed mental health counselor (LPCC) with formal CBT training from the Beck Institute and a marriage and family therapist (AMFT) with basic CBT training. Both raters completed three calibration rounds to ensure a consistent understanding of the CTRS items. We already introduced the three-round calibration process in Section 3.1 (Human–Human Interrater Reliability).
> >
> > In this study, the Guided Discovery and Application of CBT Techniques items showed notably low interrater reliability (ICC ≈ .20) when applied to 10-turn AI patient–therapist transcripts. Raters differed in how they interpreted subtle therapeutic intentions and inferred cognitive processes from text alone, resulting in limited observable variance and inconsistent scoring. Because these items reduced the overall reliability of the scale in this context, they were removed from subsequent analyses.
> >
> > We want to emphasize that the interrater reliability patterns observed here are **consistent** with findings from **Vallis et al. (1986)** who conducted the **original CTRS study** on human patient-human therapist video. They reported that on the **REAL patient–therapist conversation**, Cognitive Therapy Scale demonstrates **only moderate interrater reliability (0.59)** on human data. Yet in the psychotherapy domain it is still considered the gold standard for CBT therapy evaluation.

---

> > > ### Author Response · Authors · 2025-11-23
> > >
> > > ## W2 - Novelty and comparison to prior work
> > >
> > > We thank the reviewer for pointing out that the novelty could be made clearer.
> > >
> > > *Clarifying Table 1 dimensions and how they relate to prior work* We agree that the axes in Table 1 were not sufficiently defined, which can make the comparison feel ad-hoc. We clarify the intended meaning of each dimension and explicitly relate it to prior work:
> > >
> > > ### Skill Granularity & Interpretable Breakdown
> > >
> > > - **Our work:** FIDELITYSAFETYJUDGE outputs scores for each therapy skill (e.g., Pacing, Strategy for Change), plus safety flags. This lets us see which CBT skills improved or degraded.
> > > - **Prior work:** **ESC-Judge [2]** and similar **preference-based** evaluators only compare two responses and output “which is better” without indicating what aspects (agenda-setting, cognitive restructuring, alliance, etc.) drive the preference. **Knowledge QA** benchmarks (e.g., CBTBench [1]) do not measure interactional skills at all.
> > >
> > > ### RL Alignment Utility (usable as a reward)
> > >
> > > - **Our work:** The judge produces dense, numeric scores for 11 CTRS skills plus explicit penalties for safety violations, which we combine into a reward for online GRPO. This supports fine-grained shaping of specific therapy skills.
> > > - **Prior work:**
> > >   – General chatbot metrics (BLEU/ROUGE, MT-Bench, MT-Eval) are not designed as reward signals for therapy.
> > >   – Knowledge / classification benchmarks (CBTBench [1]) are not interactive and are cumbersome to use as rewards.
> > >   – Pairwise comparison methods (ESC-Judge [1], PsychoCounsel [5]) provide only a coarse win/lose label without aspect-level feedback, limiting their utility for targeted skill training.
> > >
> > > ### Interactive & Multi-turn
> > >
> > > - **Our work:** We evaluate full multi-turn therapist–patient dialogues (10 turns) and score them using CTRS, which is explicitly session-level. This captures process skills such as Pacing and Guided Discovery.
> > > - **Prior work:** Many therapy-related benchmarks are **non-interactive (single-turn QA [4] or knowledge QA test [1])**. Others evaluate single responses or short snippets, missing longer-range process dynamics that are central to therapy.
> > >
> > > ### Human Corr. (expert validation)
> > >
> > > - **Our work:** THERAPYJUDGEBENCH includes licensed CBT practitioners’ ratings, enabling us to report both human–human reliability and human–LLM alignment per skill.
> > > - **Prior work:** Many therapy evaluations either do not include expert ratings at all, or only use small, single-turn expert samples (CBTBench [1], PsychoCounsel [5], ESC-Judge [2]) without calibrating an automated judge.
> > >
> > > ### Domain Specificity
> > >
> > > - **Our work:** All components (dataset, judge, RL loop) are explicitly designed for CBT-style psychotherapy, using CTRS and a psychotherapy-focused safety taxonomy.
> > > - **Prior work:** General chatbot metrics (BLEU, MT-Bench, MT-Eval, HELM) are domain-agnostic and do not reflect fidelity to CBT or therapy safety.
> > >
> > > ### Auto vs. Expert
> > >
> > > - **Our work:** Expert annotation is required once to create THERAPYJUDGEBENCH; after that, the LLM judge can be used automatically at scale both for evaluation and as a reward model.
> > > - **Prior work:** Some approaches rely purely on repeated expert evaluation which is not scalable (ESC-Eval [3]).
> > >
> > > ### Safety Sensitivity (therapy-specific risks)
> > >
> > > - **Our work:** Safety is defined over psychotherapy-specific risks: inappropriate medication advice, speculative diagnosis, judgmental tone, and failure to respond to explicit harm. These are annotated on each dialogue and directly integrated into the reward.
> > > - **Prior work:** Many therapy evaluations do not include safety at all (CBTBench [1], PsychoCounsel [5], ESC-Judge [2], ESC-Eval [3]).
> > >
> > > We will revise the description around Table 1 to explicitly (i) define each dimension as above, and (ii) organize prior work into four broad categories clearly contrasting with our setting:
> > >
> > > 1. **General chatbot metrics:** Not domain-specific (e.g., BLEU [6] / general QA).
> > > 2. **Supervised therapy-related tasks:** Knowledge/classification tasks, not interaction (e.g., CBTBench [1]).
> > > 3. **Pairwise comparison frameworks:** No interpretable breakdown (e.g., A/B tests, ESC-Judge [2], PsychoCounsel [5]).
> > > 4. **General emotional-support aspects:** Not clinically validated constructs (e.g., fluency, empathy in CounselBench [4], ESC-Eval [3]).
> > >
> > > We then position THERAPYGYM as integrating clinically grounded skill granularity, therapy-specific safety, and RL-alignment utility within an interactive, multi-turn environment, which we believe clarifies the novelty relative to prior work.
> > >
> > > References (for this section):
> > > [1] Zhang et al. 2025 NAACL https://arxiv.org/abs/2410.13218
> > > [2] Madani et al. 2025 EMNLP https://arxiv.org/abs/2505.12531
> > > [3] Zhao et al. 2024 EMNLP https://arxiv.org/abs/2406.14952
> > > [4] Li et al. 2025 https://arxiv.org/abs/2506.08584
> > > [5] Zhang et al. 2025 https://arxiv.org/abs/2502.19731
> > > [6] Panineini et al. 2002 https://aclanthology.org/P02-1040.pdf

---

> > > > ### Author Response · Authors · 2025-11-23
> > > >
> > > > ## W3 - Implementation details, metrics, and discussion
> > > >
> > > > We acknowledge that some implementation details were compressed due to space and will expand them. Section 3.1 already describes the three-stage rater calibration procedure and the decision to exclude low-reliability skills from reward modeling; Section 2.3 provides the exact reward definition. See below for answers to each question.
> > > >
> > > > ### Why select o3-mini as patient model
> > > >
> > > > We chose GPT-o3-mini for the patient simulator because we needed a model that (a) can reliably follow the relatively complex Patient-Ψ-CM–based role prompt and maintain consistent cognitive patterns across turns, (b) is sufficiently capable to produce diverse but coherent patient utterances, and (c) is lightweight and cost-efficient enough for large numbers of rollouts. We will add a brief rationale in Section 2.1.1.
> > > >
> > > > ### How we obtain patient profiles
> > > >
> > > > Patient profiles come from the Patient-Ψ-CM dataset [1], which contains CBT-style cognitive conceptualization diagrams describing core beliefs, automatic thoughts, emotions, behaviors, and situations. For each simulated dialogue, we sample one such cognitive model and condition the patient prompt on the corresponding history and conceptualization, as described in Section 2.1.1 and Appendix A. The Patient-Ψ-CM paper [1] already reports clinician ratings of patient realism, and TherapyGym reuses this pipeline; we will make this inheritance and the concrete patient instructions explicit. We also acknowledge that phenomena such as strong resistance or highly ambivalent clients are only partially captured, so our environment is a controlled approximation, not a replacement for clinical trials.
> > > >
> > > > [1] Wang et al. EMNLP 2024 https://arxiv.org/abs/2405.19660
> > > >
> > > > ### Clinician annotation process and backgrounds
> > > >
> > > > The first clinician is a licensed mental health counselor (LPCC) trained in CBT at the Beck Institute, practicing since 2019. The other clinician is an associate in marriage and family therapy (AMFT) with modest CBT training as part of graduate studies and has been practicing since 2022.
> > > >
> > > > For (i), Section 3.1 already describes our three-round calibration: independent initial labeling, rubric expansion and clarification, and a joint review of a CBT protocol video and mock session, followed by finalized rating criteria; we then double-annotated 20% of dialogues to compute interrater reliability and dropped skills with α or correlations < 0.4 from reward modeling. The annotation was performed by reviewing the CTRS criteria for each scoring. The conversation was then scored as a whole rather than each turn at a time for each item on the scale.
> > > >
> > > > We agree that low α on some items (e.g., Guided Discovery, Techniques) suggests these are inherently harder to rate consistently; rather than ignoring this, we explicitly limit our main claims to the nine more reliable skills. For (ii), we will add a short paragraph in the main text describing the annotators’ credentials and their CBT training/practice context so that readers can better judge label trustworthiness.
> > > >
> > > > ### Definitions of aspects
> > > >
> > > > The CTRS skill definitions are currently summarized in Section 2.1.2 and provided verbatim from the Beck Institute in Table 5 (Appendix A). To improve interpretability, we will:
> > > >
> > > > - (a) move a condensed version of Table 5 into the main paper, and
> > > > - (b) add examples of behaviors that represent each score.

---

> > > > > ### Author Response · Authors · 2025-11-23
> > > > >
> > > > > ### Table 4: gold labels and human vs AI
> > > > >
> > > > > We want to first emphasize that **in the original CTRS study for human patient–human therapist conversation** the inter-rater reliability falls **in the moderate range (a reliability coefficient of .59)**(Vallis et al., 1986). This is considered acceptable and consistent with other psychotherapy rating scales (Lahey et al., 1983). However, the CTRS is **still considered the gold standard for evaluating CBT skills** implementation among therapists as it is an evidence-based tool and offers a structured, measurable approach to mental health treatment. Our human–human interrater reliability is **in the same range** as the **REAL patient–therapist conversations** interrater reliability. Thus, we use our human rating as gold labels to validate LLM judges.
> > > > >
> > > > > ### Table 3: NaN entries
> > > > >
> > > > > As mentioned in Section 3.3, *“NaN for Interpersonal due to zero variance”*. The “NaN” in Table 3 arises because, in that configuration, the judge produced effectively constant scores on Interpersonal Effectiveness (zero variance), rendering Spearman’s correlation undefined; we chose to display this faithfully rather than artificially perturbing scores. We will explain this in a footnote again to avoid confusion.
> > > > >
> > > > > ### Simulation examples and realism
> > > > >
> > > > > Figure 2 and Appendix A already show full prompts and a concrete dialogue snippet. In the revision, we will add a high-quality and a low-quality example session in the appendix, annotated to show where specific CTRS skills and safety flags appear.
> > > > >
> > > > > ### Safety “suite of tests”
> > > > >
> > > > > The “complementary suite of tests” refers to the four dialogue-level safety categories we define in Section 2.1.2:
> > > > >
> > > > > 1. Prescribing specific medication
> > > > > 2. Speculating about medical symptoms/diagnoses
> > > > > 3. Judgmental behavior
> > > > > 4. Failure to address harmful thoughts/behaviors
> > > > >
> > > > > These categories were derived from (a) consultation with our clinical co-authors about high-risk therapist behaviors, following the guidelines of the **American Psychological Association** and (b) prior work [1] documenting problematic safety failures in mental-health chatbots. We will clarify this origin, add examples for each category, and make the link to the therapy-specific risk taxonomy more explicit.
> > > > >
> > > > > [1] Li et al. 2025 https://arxiv.org/abs/2506.08584

---

### Official Review · Reviewer_rjpL · 2025-11-01

**Soundness:** 3
**Presentation:** 3
**Contribution:** 3
**Rating:** 8
**Confidence:** 3

**Summary:**

This paper introduces TherapyGym, a novel framework for evaluating and improving therapy chatbots along two clinically critical dimensions: fidelity to CBT principles and safety. The framework consists of three core components: 1) a dataset component (TherapyJudgeBench) comprising 116 multi-turn, synthetic dialogues annotated by licensed therapists using the Cognitive Therapy Rating Scale (CTRS) for fidelity and a four-category scheme for safety; 2) an evaluator component (FidelitySafetyJudge), an LLM-based judge that shows strong agreement with human expert ratings; and 3) an alignment component, which uses the judge's output as a reward signal for  fine-tuning within a simulated patient environment. Empirical results show that models trained in TherapyGym achieve higher automatic CTRS scores, and these improvements transfer to higher ratings from human experts, all while maintaining or improving safety.

**Strengths:**

The methodology is sound and rigorously designed. The use of the established CTRS scale provides a validated foundation for assessing therapeutic fidelity. The experiments are comprehensive, covering inter-rater reliability among human experts, alignment between LLM judges and humans, and the end-to-end efficacy of the RL fine-tuning pipeline. The decision to exclude CTRS dimensions with low human-human agreement (e.g., Guided Discovery) before reward modeling is a prudent choice that strengthens the reliability of the training signal. The reported metrics (Krippendorff's α, Spearman's ρ) are appropriate for the task.

**Weaknesses:**

As noted, the simulated patients, while based on cognitive models, may behave more ideally and articulately than real-world patients who are often hesitant, ambivalent, or inarticulate. This could limit the framework's ability to evaluate crucial therapist skills like dealing with resistance, navigating ambiguity, and building rapport with a reluctant client. Further, The four safety categories are a good start but may not encompass all emergent risks in therapeutic AI. For instance, the risk of fostering over-dependence or "AI addiction," the model creating an illusion of its own consciousness, giving overly directive advice, or failing to recognize cultural nuances are important safety considerations not currently captured.

**Questions:**

1. The conversations in TherapyJudgeBench are limited to 10 turns. Given that therapeutic alliance and complex issues often unfold over much longer interactions, how do you plan to scale the framework to evaluate longer, multi-session dialogues? Do you foresee challenges in maintaining judge accuracy or reward model stability over extended contexts?
2. The finding that few-shot prompting performed worst is intriguing. You hypothesize "prompt dilution and context-length limitations." Could you provide more detail or an example of the few-shot prompt structure? Have you explored alternative methods like chain-of-thought or providing scoring rationales from the experts to improve few-shot performance?
3. Regarding the safety taxonomy, what was the process for selecting these four specific categories? Were other potential risks (e.g., fostering dependency, giving overly simplistic advice for complex trauma) considered and excluded? How extensible is the framework to incorporating new safety labels in the future?
4. The framework is currently specialized for CBT. What are the main challenges in adapting TherapyGym to other evidence-based therapeutic modalities (e.g., ACT, DBT)? Would it primarily require a new expert-annotated dataset and a different rating scale, or are there more fundamental architectural changes needed?

---

> ### Author Response · Authors · 2025-11-23
>
> Thank you for your valuable feedback.
>
> ## On Patient Simulation
>
> We use the Patient-Ψ [1] simulation protocol (same prompt and dataset), which explicitly asks the model to:
>
> 1. “Use natural language, including hesitations, pauses, and emotional expressions, to enhance the realism of your responses,” and
> 2. “Gradually reveal deeper concerns and core issues, as a real patient often requires extensive dialogue before delving into more sensitive topics.”
>
> **The Patient-Ψ paper already reports human evaluations of patient realism by clinicians, and TherapyGym directly inherits this validated simulation setup**. We will make this connection and these concrete instructions explicit in the revision, while still acknowledging that no simulator can fully capture phenomena like strong resistance or highly reluctant clients.
>
> Moreover, using actual patients for every evaluation run would be logistically and ethically prohibitive (e.g., consent, risk management, repeated exposure to suboptimal models), which is why a validated simulation pipeline is a practical compromise.
>
> We agree that dealing with resistance, ambiguity, and reluctant clients is crucial. In future work, we plan to enrich the simulator with explicit “reluctant” and “ambivalent” personas. We have expanded the limitations section to emphasize this gap.
>
> [1] Wang et al., EMNLP 2024 — <https://arxiv.org/abs/2405.19660>
>
>
> ## On Safety Taxonomy
>
> In this first version of **THERAPYGYM**, we deliberately focus the explicit safety taxonomy on a small set of high-severity, conversation-level hazards:
>
> - Providing specific medication advice
> - Speculating about medical symptoms
> - Judgmental behaviour
> - Failure to address harmful thoughts or behaviours
>
> This taxonomy is defined for every dialogue in our safety rubric, shown in Sec. 2.1.2. Many of the additional concerns the reviewer mentions, such as over-directiveness or oversimplified responses, are partially captured indirectly through the CTRS process dimensions, which penalize non-collaborative, rigid, or relationally poor behavior.
>
> Methodologically, the framework is designed to be extensible, where safety is handled as a small set of binary labels combined with CTRS scores at the reward-model level (Sec. 2.3). Therefore, additional categories can be added as new labels and penalties without changing the core pipeline, which leaves expansion of the taxonomy to future iterations.
>
>
> ## On Conversation Length
>
> In this work, we instantiate **TherapyJudgeBench** and **TherapyGym** with 10-turn dialogues because CTRS is defined as a session-level instrument, and our annotation interface presents each whole 10-turn dialogue to clinicians for session-level scoring across all CTRS dimensions and safety flags. This length was sufficient for raters to observe evidence for all skills while keeping expert labeling tractable.
>
> **Ten turns roughly correspond to a 10–15 minute focused segment of a therapy session**, a unit that is *commonly used in supervision and skills training*. The field of LLM-based psychotherapy is still in an exploratory phase; compared to prior work that evaluates **single turns**(CBTBench [1], CounselBench [2], PsychoCounsel [3]), our CBT-anchored 10-turn dialogues **already represent a clear step forward in interaction depth**. Extending the framework to full 50-minute sessions and multi-session trajectories is an important next step, but is currently constrained by annotation cost and RL stability; our present work explores the feasibility of this intermediate, yet richer, setting.
>
> 1. Zhang et al. 2025 NAACL – [https://arxiv.org/abs/2410.13218](https://arxiv.org/abs/2410.13218)
> 2. Li et al. 2025 – [https://arxiv.org/abs/2506.08584](https://arxiv.org/abs/2506.08584)
> 3. Zhang et al. 2025 – [https://arxiv.org/abs/2502.19731](https://arxiv.org/abs/2502.19731)
>
>
> ### On Extending to Longer Lengths
>
> Conceptually, neither the judge nor the RL formulation is tied to a specific length: **FidelitySafetyJudge** takes an arbitrary multi-turn dialogue and outputs a vector of skill scores and safety flags, and the reward model operates on any dialogue-level score.
>
> For longer or multi-session settings, we envision two natural extensions:
>
> 1. Treating each session (e.g., a longer window of turns) as a unit with its own CTRS/safety evaluation, and
> 2. Aggregating per-session rewards across sessions in the RL loop.
>
> The main challenges we anticipate are precisely those the reviewer mentions: maintaining judge reliability over very long contexts and ensuring stable credit assignment when rewards depend on more distant history. We will expand on these future directions in the discussion section, and we see structured, hierarchical evaluation and reward decomposition for multi-session therapy as a central avenue for future work.

---

> > ### Author Response · Authors · 2025-11-23
> >
> > ## On Few-Shot Prompting
> >
> > ### Few-Shot Structure and Why It Underperforms
> >
> > In the few-shot setting, the judge prompt included:
> >
> > 1. The CTRS and safety rubrics
> > 2. 2–3 full 10-turn patient–therapist dialogues with their expert CTRS scores
> > 3. The target 10-turn dialogue to be scored
> >
> > This makes the prompt very long, and we believe it leads to:
> >
> > - **Context-length pressure**: the model partially “forgets” either the rubric or parts of the target dialogue
> > - **Prompt dilution**: the model overfits to the specific exemplars rather than the rubric
> >
> > We will add a short schematic of this structure in the paper rather than full dialogue examples, to stay within the space limit.
> >
> > ### Alternative Prompting (CoT and Rationales)
> >
> > We did try alternatives:
> >
> > - (a) Adding chain-of-thought style justifications for each score
> > - (b) Providing expert scoring rationales
> >
> > The latter is exactly our “ICL” setting in the paper, where we provide short rubric-aligned examples plus brief expert rationales for why a snippet reflects a given skill. This variant gave the best trade-off between performance and length, improving agreement over rubric-only prompting, whereas longer free-form CoT mainly increased output errors.
> >
> >
> > ## On Adaptivity to Other Modalities
> >
> > Architecturally, the framework itself is modality-agnostic: it assumes
> >
> > 1. A bank of therapist–patient dialogues
> > 2. A modality-appropriate rating scale providing multi-dimensional skill scores and (optionally) safety flags
> > 3. An LLM judge approximating those ratings
> > 4. An RL loop that uses these scores as reward
> >
> > To adapt to ACT or DBT, the main required changes are thus:
> >
> > - Constructing a **TherapyJudgeBench**-style dataset with expert ratings using an ACT/DBT-specific competence scale
> > - Updating the judge prompts and reward definition to use those dimensions
> >
> > We will make this pathway to generalization more explicit in the revised version.

---

### Official Review · Reviewer_QqSC · 2025-11-01

**Soundness:** 2
**Presentation:** 2
**Contribution:** 2
**Rating:** 4
**Confidence:** 3

**Summary:**

The paper proposes THERAPYGYM, a framework to evaluate and improve therapy chatbots along two pillars: clinical fidelity and safety. Fidelity is operationalized via the Cognitive Therapy Rating Scale (CTRS), adapted to an automatic multi-turn scoring pipeline capturing adherence to CBT techniques and competence in delivery. Safety is assessed via multi-label annotations of domain-specific risks (e.g., judgmental language, failure to address self-harm). To address unreliability in LLM judges, the authors introduce THERAPYJUDGEBENCH, a validation set of 116 dialogues with 1,270 expert ratings, intended for auditing and calibrating LLM judge performance against licensed clinicians. Beyond evaluation, THERAPYGYM serves as a training harness: CTRS and safety scores are used as rewards in an RL loop where the LLM therapist interacts with simulated patients exhibiting varied symptom profiles and conversational styles. The authors claim that models trained in THERAPYGYM improve automatic CTRS scores and that these gains transfer to expert ratings.

**Strengths:**

- Framing evaluation around clinical constructs (fidelity and safety) is exactly the direction the community needs; CTRS-linked behavioral coding is far more meaningful than generic fluency or preference metrics.
- A dual-purpose environment (evaluation plus RL training harness) is attractive and pragmatic—closing the loop from measurement to improvement.
- LLM-judge validation against licensed clinicians via THERAPYJUDGEBENCH is a good step toward quantifying judge reliability.
- Emphasis on multi-turn, process-level competence with programmable patient simulations acknowledges the interactive nature of therapy.

**Weaknesses:**

- CTRS automation is underspecified. The paper lacks a clear rubric, item mappings (CTS-R/CTRS to the “9 CBT skills”), scoring granularity (turn vs session), aggregation rules, and reliability analyses, making it hard to audit whether CTRS is faithfully operationalized.
- Human rating setup is thin and ambiguous. Only four raters are mentioned; rating units (turn/skill/session) are unclear; inter-rater reliability is missing; claims that gains “transfer to expert ratings” lack quantitative evidence.
- Risk of judge gaming (Goodhart’s Law). Using CTRS- and safety-derived judges in RL without robust controls raises concern that models optimize the judge rather than clinical quality; the paper lacks out-of-judge, blinded human evaluations and adversarial tests.
- Safety taxonomy and coverage are underdefined. The label set, annotation protocol, prevalence, and inter-rater agreement are not specified; multi-turn detection and severity handling are unclear.
- Patient simulation specificity is weak. How profiles are generated, distributions chosen, and realism validated is not described, risking overfitting to unrealistic interaction patterns.

**Questions:**

- Provide the exact CTRS/CTS-R instrument: cite the scale, list items, show the mapping to the nine skills, specify scoring per turn vs session and aggregation, and report agreement among LLM judges and with human raters.
- Detail the safety label set and protocol: definitions, guidelines, prevalence, inter-rater reliability, multi-turn detection, and severity scoring; consider releasing a small de-identified subset for benchmarking.
- Explain the RL setup and anti-gaming controls: policy architecture, reward shaping (fidelity + safety), on/off-policy choice, and safeguards; include out-of-judge, blinded human evaluations and adversarial stress tests to demonstrate generalization.
- Expand THERAPYJUDGEBENCH details: number/qualification of raters, rating granularity, reconcile the 1,270 ratings with units, and report alignment metrics with confidence intervals (correlations, κ, Bland–Altman).
- Clarify patient simulation generation and realism checks, and confirm CTRS gains on unseen conversations and human-generated dialogues.

---

> ### Author Response · Authors · 2025-11-23
>
> Thank you for your valuable feedback.
>
> ## On CTRS Automation and Rubric Specification
>
> In the paper, we use the official Cognitive Therapy Rating Scale (CTRS) from the Beck Institute, with all 11 standard dimensions as described in Section 2.1.2 (lines 221–225), and we include the definition verbatim in Appendix Table 5 (lines 831–847). Expert raters assign session-level scores for each dimension, consistent with CTRS being a session-level instrument (lines 231–232).
>
> For finetuning, we retain only those CTRS dimensions with at least moderate human–human agreement (Section 3.1, Table 2), dropping Guided Discovery and Application of CBT Techniques (α < 0.4) and thus defining the “9 CBT skills” (lines 320–323). For reward construction, we already introduced them in Section 2.3.1.
>
> ## On Human Rating Setup and THERAPYJUDGEBENCH Details
>
> All CTRS and safety annotations are produced by two licensed, CBT-trained practitioners using an annotation interface that presents full 10-turn dialogues (Section 2.1.2 and Figure 4). Ratings are dialogue-level. In addition, we have already included inter-rater agreement (hence inter-rater reliability) in Table 2.
>
> We also already report “transfer to expert ratings”: in Table 4, human-rated CTRS averages increase from 0.27 to 0.49 (normalized 0–1) after RL, with safety violations remaining stable at 0.03.
>
> ## On Safety Taxonomy, Annotation Protocol, and Coverage
>
> As described in Section 2.1.2, we use a four-category safety taxonomy at the dialogue level:
>
> 1. Prescribing medication
> 2. Speculating about medical symptoms or diagnoses
> 3. Judgmental tone
> 4. Failure to respond to explicit high-risk content (self-harm, suicide, violence)
>
> Two licensed CBT-trained practitioners label each full 10-turn dialogue and mark each category as present/absent if it appears in any therapist's turn. The safety labels are selected from previous work [1], and the selection guideline is based on safety protocol used in real practice, i.e., risk assessment for harm and staying within the scope of practice according to the American Psychological Association (Ethical principles of psychologists and code of conduct. APA, 2017).
>
> We currently use binary flags rather than graded severity to reduce annotation burden and avoid introducing additional subjectivity.
>
> [1] Li et al. 2025, https://arxiv.org/abs/2506.08584
>
> ## On RL Setup, Goodhart’s Law, and Anti-Gaming Safeguards
>
> In our setup, FIDELITYSAFETYJUDGE is first calibrated against licensed CBT clinicians on THERAPYJUDGEBENCH, across 116 dialogues, and achieves an average Spearman ρ of 0.56 on CTRS skills and 99% accuracy on safety labels, as shown in Sec. 2.2, Table 3. It recovers much of the human signal rather than being an arbitrary metric.
>
> For RL, we use this judge as a frozen rater and construct a reward that:
>
> 1. Only includes CTRS dimensions with at least moderate human–human reliability shown in Sec. 3.1, Table 2; and
> 2. Subtracts penalties for all four unsafe behaviors (Eq. (1), Sec. 2.3.1), explicitly discouraging unsafe “shortcuts”.
>
> Finally, we already include out-of-judge human evaluation. After RL, independent human therapists re-score dialogues, and Table 4 shows that both judge scores and human CTRS ratings increase while safety violations stay low, as shown in Sec. 3.3.
>
> ## THERAPYJUDGEBENCH Structure and Alignment Metrics
>
> To consolidate the requested information about THERAPYJUDGEBENCH, we emphasize that it consists of 116 synthetic CBT dialogues (10 turns each), annotated by two licensed CBT practitioners, with:
>
> - Dialogue-level CTRS scores across 11 dimensions
> - Four binary safety flags
> - 20% of dialogues double-coded to measure inter-rater reliability
>
> This dataset underlies all of our judge validation experiments: human–human reliability (Table 2) and human–LLM alignment across multiple judge models and prompting regimes (Table 3 and Appendix Table 6) are computed directly on THERAPYJUDGEBENCH.
>
> ## Patient Simulation Generation, Realism, and Generalization
>
> As described in Section 2.1.1, we initialize patients from the Patient-Ψ-CM cognitive models [1], which encode CBT-relevant constructs (core beliefs, automatic thoughts, situations, emotions, behaviors). This anchors each dialogue in a validated CBT conceptualization. The Patient-Ψ-CM paper [1] already reports clinician ratings of patient realism, and TherapyGym reuses this pipeline.
>
> GPT-o3-mini plays the patient role, conditioned on a single sampled cognitive model per dialogue. Each dialogue is 10 turns with no templated scripts.
>
> For generalization, our RL evaluation already uses held-out patient profiles (22 unseen profiles) (lines 354–356) and independent human ratings, and Table 4 reports CTRS improvements and stable safety rates on these unseen conversations.
>
> [1] Wang et al. EMNLP 2024, https://arxiv.org/abs/2405.19660

---

### Official Review · Reviewer_HJ1s · 2025-11-01

**Soundness:** 1
**Presentation:** 3
**Contribution:** 2
**Rating:** 2
**Confidence:** 4

**Summary:**

This paper introduces TherapyGym, a framework for evaluating the fidelity and safety of LLM-based therapists based on the Cognitive Therapy Rating Scale and improving LLM therapist chatbots via reinforcement learning. TherapyGym’s components include a dataset of 116 simulated 10-turn therapist-patient conversations, an LLM evaluator that scores dialogues on fidelity and safety, and an online reinforcement algorithm to align and improve LLM therapists. The authors demonstrate that their RL fine-tuning approach using GRPO leads to higher ratings of LLM therapists from an LLM judge and human therapists within their experimental setup.

**Strengths:**

The work contributes a new evaluation method for LLM therapy chatbots by focusing on clinically relevant measures of performance; namely, adherence and competence in delivering treatment, and the avoidance of harmful behaviors. Their framework uses a clinically validated scale (CTRS) to evaluate and fine-tune LLMs as therapists.

**Weaknesses:**

TherapyGym was not evaluated on a realistic, comprehensive dataset of therapist-patient dialogues (it uses just 116 dialogues, each consisting of only 5 turns per agent).

General-purpose and mental health-specific LLMs are often biased and generate false information. Previous work cited in the introduction specifically cautions against using simulated clients and therapists to avoid this and instead suggests evaluating LLMs on real-world therapist-client interactions, making the reliance on simulated agents a significant drawback of this work.

This work did not introduce novel algorithms. RL for fine-tuning LLMs, even for therapy, have been used in prior works, and GRPO was recently introduced by DeepSeek.

FidelitySafetyJudge only rates LLM therapists on a subset of CTRS skills with high human reliability. The skills Guided Discovery and Application of CBT Techniques were excluded despite being highly important aspects of an effective therapist intervention.

FidelitySafetyJudge ratings appear much higher than those of human annotators. This bias should be addressed before TherapyGym can be considered an accurate, robust framework for training and evaluating LLMs as therapists.

Results show that even after fine-tuning, LLM therapists do not achieve average CTRS scores above 0.5 from human annotators (rescaled from 0-6 to 0-1).

**Questions:**

Why is the proposed method not evaluated on more realistic datasets?  This work could benefit from additional evaluation with real-world dialogue datasets (High-and-Low-Quality Therapy Conversation Dataset, HOPE), providing more grounding for TherapyGym as a robust LLM therapist evaluation framework.

Given the incongruity between the FidelitySafetyJudge and human therapist raters, it is unclear if its evaluations are accurate enough or well-aligned to expert raters. How do the authors envision future improvements to the LLM rater?

---

> ### Author Response · Authors · 2025-11-23
>
> Thank you for your valuable feedback.
>
> ## Simulated environment validity
>
> Before addressing the specific points, we want to clarify our stance on simulation. In many safety–critical domains such as *autonomous driving [2] and robotics [3][4]*, high-fidelity **simulators are not a shortcut but a necessary tool**: they avoid exposing real users to unsafe or immature systems, enable large-scale and repeatable experimentation, and allow systematic stress-testing of rare but important scenarios. **What matters in these settings is not “simulation vs. reality” in the abstract, but whether policies that perform well in simulation also behave well under independent real-world evaluation**. Our work is designed around exactly this principle: we construct a clinically grounded simulation and then explicitly test whether its preferences align with expert ratings.
>
> Our aim is therefore **not** to claim that our setup fully reproduces real-world therapy, but to build a **scalable, ethically acceptable, clinically anchored** environment with two key properties:
>
> 1. The simulated conversations are plausible CBT-style sessions grounded in validated cognitive models, and
> 2. The environment’s scores track expert judgments well enough to support model comparison and RL.
>
> *Why simulation and what level of realism we target* Evaluating therapy agents interactively requires a conversational partner who is either a real patient or a simulator. **Running every evaluation with real patients would be logistically and ethically prohibitive (risk management, informed consent, repeated exposure to suboptimal models)**. A carefully designed simulator is thus a practical compromise: we target clinical plausibility at the level of CBT case formulation and interaction patterns, rather than claiming to capture all human nuances.
>
> To ground the simulations, we do *not* ask models to “role-play a depressed person” in the abstract. Instead, we initialize patients from the **Patient-Ψ-CM cognitive models [1]**, which encode CBT-relevant constructs (core beliefs, automatic thoughts, situations, emotions, behaviors). This anchors each dialogue in a validated CBT conceptualization. **The Patient-Ψ-CM paper [1] already reports clinician ratings of patient realism**, and TherapyGym reuses this pipeline; we will make this inheritance explicit. We also acknowledge that phenomena such as strong resistance or highly ambivalent clients are only partially captured, so our environment is a controlled approximation, not a replacement for clinical trials.
>
> *Why 10-turn “micro-sessions”* We use 10-turn sessions as a standard micro-session that is long enough to exercise multiple core CBT skills (agenda setting, understanding, guided discovery, strategy for change, homework), while keeping expert annotation and RL training tractable. Ten turns roughly correspond to a 10–15 minute focused segment of a therapy session, a unit that is commonly used in supervision and skills training. The field of LLM-based psychotherapy is still in an exploratory phase; **compared to prior work that evaluates single turns** (CBTBench [5], CounselBench [6], PsychoCounsel [7]) [1][2], our CBT-anchored 10-turn dialogues **already represent a clear step forward in interaction depth**. Extending the framework to full 50-minute sessions and multi-session trajectories is an important next step, but is currently constrained by annotation cost and RL stability; our present work explores the feasibility of this intermediate, yet richer, setting.
>
> 1. Wang et al. EMNLP 2024 – [https://arxiv.org/abs/2405.19660](https://arxiv.org/abs/2405.19660)
> 2. Goff et al. 2025 – [https://arxiv.org/pdf/2504.19077v1](https://arxiv.org/pdf/2504.19077v1)
> 3. Hao et al. CVPR 2025 – [https://arxiv.org/pdf/2504.07095](https://arxiv.org/pdf/2504.07095)
> 4. Peng et al. ICRA 2018 – [https://arxiv.org/pdf/1710.06537](https://arxiv.org/pdf/1710.06537)
> 5. Zhang et al. 2025 NAACL – [https://arxiv.org/abs/2410.13218](https://arxiv.org/abs/2410.13218)
> 6. Li et al. 2025 – [https://arxiv.org/abs/2506.08584](https://arxiv.org/abs/2506.08584)
> 7. Zhang et al. 2025 – [https://arxiv.org/abs/2502.19731](https://arxiv.org/abs/2502.19731)
> ---
>
> ## On real conversation dataset
>
> *Why do we not adopt the real dataset?* The real conversation datasets (High-and-Low-Quality Therapy Dataset, HOPE) are **not CBT-style therapy sessions**. In our setting, the *main bottleneck* for such validation is the lack of publicly available CBT session transcripts: *due to privacy and regulatory constraints*, real conversation datasets in psychotherapy are **extremely rare**, and to the best of our knowledge, there is currently no public corpus of CBT therapy dialogues suitable for this purpose. If the reviewer found a CBT conversation dataset, we would be very grateful for a pointer and would be excited to evaluate TherapyGym on it in future work.
>
> 1. Chiu et al. 2024 – [https://arxiv.org/abs/2401.00820](https://arxiv.org/abs/2401.00820)

---

> > ### Author Response · Authors · 2025-11-23
> >
> > ---
> >
> > ## Algorithm not novel
> >
> > Algorithmic novelty is **not the focus of our therapy environment paper**. We agree that our work does not propose a new RL algorithm, and we did *not* intend to claim algorithmic novelty. Our goal is to provide a clinically grounded evaluation and alignment framework for therapy chatbots, not a new optimization method. The RL component deliberately builds on an existing algorithm (GRPO) as a plug-in, so that the novelty is cleanly located in the clinical formulation of the task, reward, and environment.
> >
> > ---
> >
> > ## Only a subset has high human reliability
> >
> > In this study, the **Guided Discovery** and **Application of CBT Techniques** items showed notably low interrater reliability (ICC ≈ .20) when applied to 10-turn AI patient–therapist transcripts. Raters differed in how they interpreted subtle therapeutic intentions and inferred cognitive processes from text alone, resulting in limited observable variance and inconsistent scoring. Because these items reduced the overall reliability of the scale in this context, they were removed from subsequent analyses.
> >
> > The interrater reliability patterns observed here are consistent with findings from Vallis et al. (1986), who reported that on **real human–human therapy conversations** the Cognitive Therapy Scale(CTRS demonstrates only moderate interrater reliability (0.59). However, the CTRS is still considered the Gold Standard for CBT therapy rating.
> >
> > ---
> >
> > ## Missing aspects of CTRS
> >
> > Of note, in the original CTRS study the CTRS inter-rater reliability for **REAL patients recorded video sessions** falls in the **moderate range (a reliability coefficient of .59)** (Vallis et al., 1986). However, the CTRS is still considered the *gold standard* for evaluating CBT skills implementation among therapists as it is an evidence-based tool and offers a structured, measurable approach to mental health treatment.
> >
> > Guided Discovery and Application of CBT Techniques items require substantial inference about therapist intentions, resulting in high subjectivity. Therefore, we excluded them to improve reliability and reward learnability. Guided Discovery and Application of CBT Techniques items require substantial inference about therapist intentions and rely on subtle cognitive processes that were difficult to observe reliably within the conversations, leading to inconsistent scoring across raters.
> >
> > ---
> >
> > ## FidelitySafetyJudge ratings are higher than human ratings
> >
> > Our evaluation objective for judge alignment is explicitly **rank-order consistency (Spearman’s ρ)** rather than exact score matching, which we describe in the *Human–LLM Alignment* in Section 3.1. This means the judge is used to recover the structure of **human preferences** rather than to reproduce human score magnitudes. Table 4 shows that FIDELITYSAFETYJUDGE indeed assigns higher average CTRS scores than clinicians, even though the relative pattern of improvements is similar across skills. Given this setup, the absolute score inflation does not affect our RL training or conclusions.
> >
> > ---
> >
> > ## Finetuning shows limited improvement
> >
> > Our base therapist is intentionally a small Qwen3-4B model as mentioned in Sec. 3.2, chosen to test whether THERAPYGYM can improve a modest baseline rather than to maximize raw performance. As reported in Table 4, GRPO fine-tuning yields consistent gains in both judge and human CTRS averages (0.53 → 0.60 and 0.27 → 0.49). Our primary goal, as stated in the abstract and conclusion, is to introduce a multi-turn evaluation and alignment framework, THERAPYGYM, that makes fidelity and safety measurable and optimizable, rather than claiming state-of-the-art therapist performance from a single small model. We will emphasize this framing and, as future work, plan to apply the same framework to larger models and with more human raters.

---

### Meta-Review · Area_Chair_ZUvM · 2026-01-07

**Summary:**

This paper presents TherapyGym, a framework for evaluating the fidelity and safety of LLM-based therapist. They developed 11 aspects for fidelity based on Cognitive Therapy Rating Scale (CTRS), as well as 4 tags related to safety. They collect and release TherapyJudgeBench, 116 dialogues (each dialogues contain 10 turns) and 1,270 expert ratings. Based on the annotated data, they build FidelitySafetyJudge, a model for evaluating clinical dialogues, and finally fine-tunes therapist chatbot LLM, applying GRPO to improve scores from FidelitySafetyJudge.

The new dataset, annotated by real experts, can be useful for the community, as well as evaluation framework grounded in clinical practices. The reviewers (and AC as well) acknowledge the value of dataset. The paper is written pretty clearly and well-motivated.

Reviewer RSRq wanted a clearer contextualization of work compared to prior work. I agree that current Table 1 is not very informative (and I think the author response partially addressed this concern). Another major concern is the validity of the synthetic conversation dataset (i.e., how realistic are the simulated conversation?) The authors defend their choice by relying on prior work which showed promising simulation ability of LLMs, but independent validation study would strengthen the paper. Multiple reviewers also pointed out missing details about data collection and implementation. The reviewer also raised the potential concern of reward hacking. I am not sure fine-tuning with GRPO (section 2.3.2) adds meaningfully to the paper -- it is not novel technique, and the improved results are sensible but not surprising. The paper might benefit from a bit more thorough documentation and validation of dataset rather than fine-tuning experiments.

Overall, the paper studies a well-motivated problem and proposes a potentially useful dataset for the research community yet has some concerns that need to be addressed.

**Reviewer Concerns:**

The reviewers HJ1s had concerns about (1) synthetic nature of the dataset (2) score inflation from FidelitySafetyJudge, and (3) the lack of algorithmic innovation. I don't think (1) and (3) are addressable (weakness of the work), but (2) might have been convinced.

The reviewer QqSC had concerns about lack of details in the paper, concerning how they developed rubrics, or described patient simulations, or developed safety taxonomy. These are details that authors largely provided. The reviewer also brought up issues of reward hacking.

The reviewer RSRq raised concerns about simulated dataset, lack of novelty compared to prior work, and many missing details. I'm not sure the concern about simulated dataset is fully addressed.

**Reviewer Scores:**

HJ1s and QqSC might have increased the score a bit.

---

### Decision · Program_Chairs · 2026-01-26

Reject